## Research Article

coastal geomorphology; morphodynamics; shoreline change; cyclone; remote sensing

**Corresponding author:**
Holly Muecke;
Email: hqwm045@uowmail.edu.au

# Effects of tropical cyclone Jasper at Low Island, northern Great Barrier Reef

Holly Muecke[1,2] ⬛, Scott Smithers[1], Stephanie Duce[1], Sarah Hamylton[2] ⬛ and Emily Lazarus[1]

[1]Earth and Environmental Sciences, College of Science and Engineering, James Cook University, Townsville, QLD, Australia and [2]Environmental Futures Research Centre, School of Science, University of Wollongong, Wollongong, NSW, Australia

## Abstract

Tropical cyclones (TC) can produce waves and water levels that markedly reshape sand cay shorelines. TC Jasper (December 2023) passed near Low Island (Low Isles, Northern Great Barrier Reef [GBF]) as a category 2 storm. Using a combination of remote sensing and ground surveys, we compare detailed, high-resolution digital terrain models created before and after TC Jasper to quantify sediment redistribution around the cay during and after the event. During TC Jasper, net transport of 8,870 m³ occurred to elongate the spits at the eastern and western ends of the cay, but the sediment volume of the cay did not significantly change. Following TC Jasper, the shoreline at Low Island returned to its modal seasonal state within six months. This accords with historical accounts of seasonal shifts in shoreline configuration driven by prevailing wind and wave regimes, as well as the relatively rapid readjustment to a modal form following episodic extreme events. Overall, the documented changes to Low Island following cyclonic events highlight the complex interplay between episodic disturbances and longer-term geomorphic stability, emphasising the importance of ongoing research into these interactions as higher-intensity cyclones increase in frequency due to climate change.

## Impact statement

Reef islands are globally recognised as vulnerable to climate change, particularly due to the increasing frequency and intensity of storms and TCs (Kennedy, 2024). TCs can cause major changes to reef islands, including major shoreline reconfiguration due to large-scale sediment redistribution (Perry et al., 2014; Smithers and Hoeke, 2014). Remote sensing and three-dimensional modelling applied in this study enable detailed assessment and quantification of shoreline changes resulting from TC Jasper – a moderately sized category 2 system active in the region between 2nd–13th December 2023. Reef islands are largely composed of unconsolidated biogenic carbonate sediments produced on the surrounding reef. Reef island shoreline changes have a dynamic complexity that cannot be reduced to simple net erosion or accretion; rather, they exhibit spatial and temporal patterns of localised shoreline change, including vertical adjustments (Kench and Brander, 2006; Webb and Kench, 2010). This study is focussed on the Low Isles, but the findings are relevant to cays globally. The results document the distribution and magnitude of shoreline changes during and after a modest cyclone event. They also provide insights into potential reef island dynamics under climate-change-modified storm regimes, offering information useful to reef island communities and ecosystem managers.

## Introduction

Reef islands are low-lying sedimentary landforms that are largely composed of unconsolidated biogenic carbonate sediments produced by organisms on the host reef platform (Gourlay, 1988; Kench and Brander, 2006; Webb and Kench, 2010; Bonesso et al., 2020). Their formation and stability reflect the interplay of local geomorphic and environmental processes operating across time scales – from relative sea-level change over centuries to millennia to daily – decadal hydrodynamic variability, including episodic high-magnitude storms (Hamylton et al., 2019). These processes are increasingly affected by anthropogenic climate change and sea-level rise, threatening both reef islands and dependent ecosystems and communities (East et al., 2018; Shope and Storlazzi, 2019; Matera, 2020; Kench et al., 2023; Steibl et al., 2024). This study documents shoreline dynamics of Low Island, a sandy cay on the Great Barrier Reef (GBF), during and immediately following the passage of tropical cyclone (TC) Jasper in December 2023. Detailed surveys prior to and following the event enable quantification of the impact of such storms on the morphology of small unconsolidated reef islands, including post-event adjustments.

TCs are rotating low-pressure systems that form between 7° and 25° north and south of the equator over oceans warmer than 27 °C (Perry et al., 2014; Geoscience Australia, 2022). TCs are associated with sustained gale force winds (>64 km/h) gusting up to 300 km/h, heavy rainfall, significantly increased wave heights and storm surges exceeding normal tide levels (Perry et al., 2014; Geoscience Australia, 2022; BOM, 2024). Although cyclones are short lived, the elevated water levels and waves they generate may exceed seasonal extremes and significantly modify the geomorphology of reefs and reef islands (Stoddart, 1962; Scoffin, 1993; Perry et al., 2014; Smithers and Hoeke, 2014; Ford and Kench, 2016). Destructive impacts include stripping of live coral cover, damage to reef structures, and burial of living reef areas with rubble and sand (Harmelin-Vivien, 1994; Puotinen, 2007; Jones et al., 2019), while constructive effects include sediment production and accumulation of rubble banks and ramparts (Maragos et al., 1973). Early work on the effects of cyclones on reef islands – including studies in Jaluit Atoll in the Marshall Islands (Blumenstock, 1958), Belize (formerly British Honduras) (Stoddart, 1971), Ontong Java Atoll in the Solomon Islands (Bayliss-Smith, 1988) and Funafuti Atoll in Tuvalu (Maragos et al., 1973; Baines and McLean, 1976a, 1976b) – highlighted the complex and locally variable nature of reef island responses. Stoddart (1971) identified patterns of zonal damage, greater in smaller, narrow islands near the storm centre, causing marginal erosion, vegetation loss, and the disappearance of some cays. Bayliss-Smith (1988) underscored the cyclical formation and destruction of cays, noting that some can reform after cyclone-driven erosion if sediment supply and transport conditions are favourable. Work on Funafuti Atoll similarly demonstrated how cyclones can generate entirely new depositional features and how storm ramparts evolve through time (Maragos et al., 1973; Baines and McLean, 1976a, 1976b). Collectively, these studies show that high-energy cyclones can reshape reef islands. Waves and currents deliver coarse sediment onto the reef flat, where it is reworked into sand over time and may also deposit windward shingle ramparts that create sheltered low-energy zones on the reef platform (Smithers and Hoeke, 2014; Maragos et al., 1973; Baines and McLean, 1976; Mann and Westphal, 2016).

Despite these early studies, significant knowledge gaps remain regarding how cyclones impact cays. Detailed high-resolution 3D datasets for cays captured within a few weeks prior to and following a cyclone, from which sediment movements can be accurately measured, are rare. As climate change modifies storm regimes, it is critical to address these uncertainties , particularly when informing management strategies (Dawson, 2021). More systematic and technologically advanced monitoring efforts to quantify shoreline changes and sediment dynamics, both in terms of short-term cyclone damage and long-term island resilience, are urgently needed.

Cyclones on the GBR usually occur during the warmer months between November to April (BOM, 2024), with the greatest number typically forming during La Nina years when Coral Sea sea-surface temperatures are relatively warm (BOM, 2024). This study investigates the geomorphic impacts of TC Jasper on the shoreline morphology of Low Island, a small, vegetated sand cay located on the inner GBR shelf approximately 60 km north of Cairns. Detailed observations at Low Island, alongside historical records of seasonal and cyclone impacts dating back to the first Great Barrier Reef Expedition, offer a unique opportunity to understand processes shaping long-term shoreline dynamics on small dynamic cays globally (Moorehouse, 1936; Fairbridge and Teichert, 1948;

Hamylton et al., 2019; Spencer et al., 2021). The geomorphic impacts were established by:

1.  generating 3D elevation models of Low Island prior to and following TC Jasper using remote sensing and ground survey data to quantify geomorphological changes caused by the cyclone.
2.  comparing shoreline changes associated with TC Jasper to those produced by the historic 1934 cyclone and longer-term seasonal shoreline dynamics to better understand the relative significance of episodic high-energy events compared to shoreline adjustments driven by seasonal wave regimes

### Study site

Low Isles (16.3840° S, 145.5599° E) is located 15 km northeast of Port Douglas (Figure 1). It is a low wooded-type reef island (sensu Hopley, 2011; Hopley et al., 2007) comprising two geomorphic units: Low Island, a vegetated sand cay (~250 m long and 100 m wide) deposited on the north-western corner of the reef platform, and Woody Island, which consists of shingle ridges and a leeward mangrove forest located on the windward east to southeastern reef margin. This study is focussed on Low Island.

Low Isles is culturally important to the Traditional Custodians, the Kuku Yalanji and Yirraganydji people, who know it as Wung-kun. European heritage assets are also located at Low Isles, including the lighthouse built in 1874 (GBRMPA, 2018). Low Isles is historically significant as the site of the first scientific expedition on the GBR, the 1928–1929 Great Barrier Reef Expedition, and its subsequent status as the site for much ongoing reef and reef island research (see review by Spencer et al., 2021). A long history of detailed records of geomorphic changes at Low Island offers an unparalleled opportunity to understand geomorphic changes to the reef top landforms, including the mangrove forest, sand cay and shingle spits over extended timeframes. Low Isles also supports a range of critical ecological values including as a vital roosting site for endangered pied imperial pigeons and other seabirds (Brothers and Bone, 2012; Hamylton et al., 2019). Hundreds of tourists visit Low Isles on most days, attracted by these cultural, ecological and historical values.

### Low island

Low Island is an oval-shaped vegetated sand cay with its long axis aligned east–west and mobile sandy spits at its eastern and western ends. The northern and southern shorelines include beachrock that is periodically buried and exposed (Stephenson et al., 1958; Stoddart et al., 1978; Frank and Jell, 2006). The cay is predominantly composed of medium-to-coarse coral, mollusc and foraminifera sands (Fairbridge and Teichert, 1948). The vegetated area of the cay is mostly 4–5.5 m above lowest astronomical tide (LAT), with the highest areas along the western and southern margins (Hamylton et al., 2019). Low Island's vegetation has been recently described as dense or "well wooded", including species such as *Casuarina, Tournefortia, Scaevola and Ipomoea* (Hamylton et al., 2019; Stoddart et al., 1978). However, Moorehouse (1936) records the destruction of the cay's vegetation during a cyclone in 1934, and photographs from 1945 show sparser vegetation than present (Fairbridge and Teichert, 1948). Clearing for building construction and exotic plantings (e.g. frangipanis, hibiscus) has also modified vegetation cover.

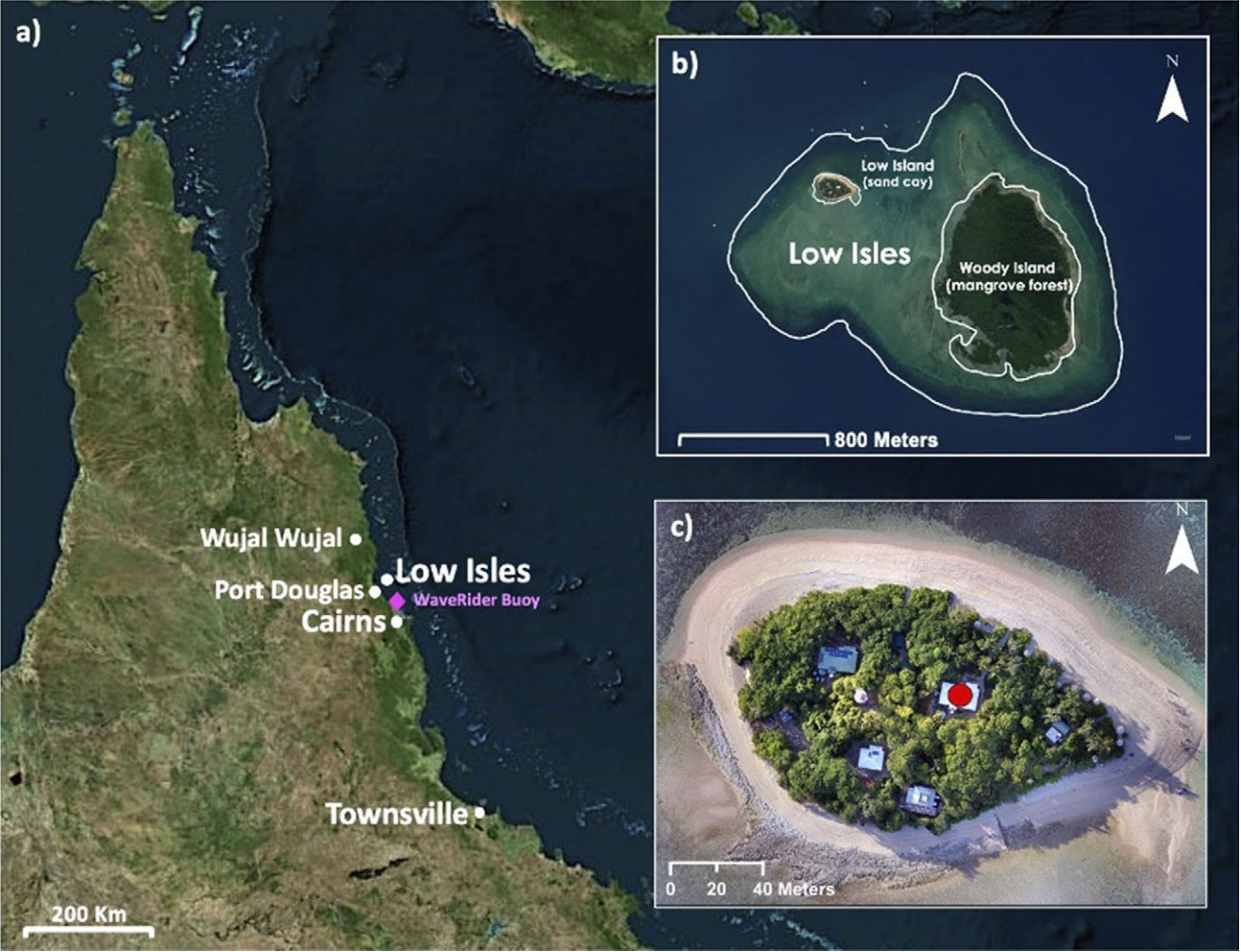

**Figure 1.** (a) Location of Low Isles in relation to Cairns, Townsville, Port Douglas and Wujal. Location of the Cairns WaveRider buoy marked in pink. (b) Low Isles reef platform showing the location of Low and Woody Islands. (c) Low Island captured in June 2023; the Caretaker's residence is marked with a red circle.

## Hydrodynamic regime

The hydrodynamic regime at Low Isles is largely a function of seasonal patterns. Southeasterly trade winds dominate from May to September. From October to April, the northwest monsoon brings heavier rainfall, weaker, more variable winds and episodic TCs. The wave climate at Low Isles is dominated by locally generated wind-driven waves as the outer GBR reef tract attenuates up to 95% of swell waves propagating from the Coral Sea (Gallop et al., 2014). Waves therefore predominantly approach from the southeast and at higher tides propagate across the reef platform towards the northwest (Frank, 2008). The closest WaveRider buoy, approximately 39 km to the south (16°43.830′S, 145°42.910′E, Figure 1), has collected wave data since 1975 (https://www.qld.gov.au/environment/coasts-waterways/beach/monitoring/waves-sites/cairns), establishing that the wave climate at Low Isles is seasonally variable. Short period (mean Tp 5.6 s) and generally larger waves (mean Hs 0.52 m, Hmax 2.0 m) prevail between May and September, whereas slightly smaller (mean Hs 0.46 m) and shorter period (Tp 4.7 s) are more typical during the NW monsoon. Importantly, however, NW monsoon conditions are more variable, with periods of much lower wave energy associated with wet season doldrums and episodic storms generating waves that may exceed 3 m. During the dry season, waves reliably approach Low Isles from the east-southeast in accord with the prevailing southeasterly trade winds, while storm waves in the wet season may approach from a broad range of directions depending on storm track geometry. The tides at Low Isles are semi-diurnal, with the highest astronomical tide (HAT) being 3.41 m above LAT (Queensland Government, 2025).

## Tropical cyclone Jasper 2023

Severe TC Jasper (2–13 December 2023) peaked as a category 5 cyclone in the Coral Sea, south of the Solomon Islands, before weakening to category 1 as it approached the GBR (Figure 2) (Prasad, 2024). It then moved westwards, reintensifying into a category 2 cyclone before making landfall on the north Queensland coast near the township of Wujal, 57 km north-northwest of Low Isles, on 13 December (8 pm AEST, UTC + 10 h). The strongest winds, with gusts up to 130 km/h, were recorded south of the system, extending to Port Douglas and Low Isles. The cyclone rapidly weakened after landfall and was downgraded below TC intensity by midnight on 14 December. TC Jasper caused widespread impacts along the Far North Queensland coast due to heavy rainfall, high winds and storm waves (BOM, 2024).

Sustained winds above 89 km/h were recorded at Low Isles between 1:30 pm and 4:10 pm on December 13, with gusts peaking

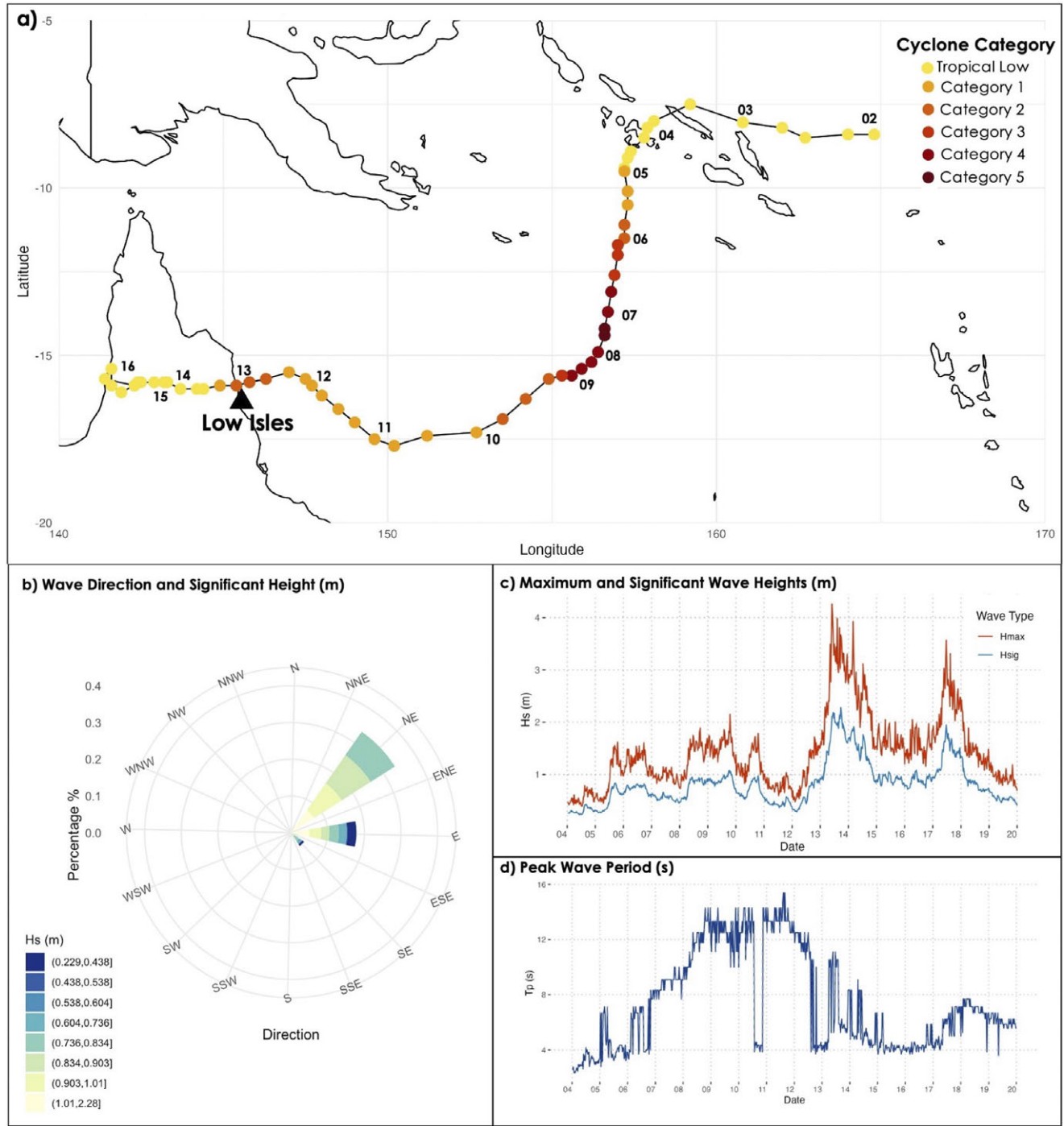

**Figure 2.** (a) Track of severe tropical cyclone Jasper in relation to Low Isles (black triangle), with dates labelled from 02 to 16 December 2023. The colours of the points represent the cyclone category, determined based on central pressure (hPa). As the cyclone passed its closest point to Low Isles (some 57 km to the northwest), it was rated category 2, which equates to sustained winds of 89–117 km/h. Wave climate measured at the Cairns WaveRider buoy from 4th–20th December 2023; (b) wave direction and significant wave height (m), (c) maximum and significant wave heights (m) and (d) peak wave period (s). All data downloaded from the Bureau of Meteorology (2023).

at 115 km/h at 3:30 pm (Prasad, 2024). At peak high tide, water levels at Low Isles were approximately 0.4 m above the predicted peak tide level (2.82 m LAT). At high tide on December 13, significant wave heights at the Cairns waverider buoy averaged 2.2 m, and the largest waves exceeded 4 m (Hmax).

As TC Jasper tracked north of Low Isles, the largest waves approached from the northeast (Figure 2). These waves propagated towards the northern shoreline of Low Island, unobstructed by the shingle ramparts and mangroves on the reef platform margin to the east and southeast. Significant wave heights above 1 m and maximum wave heights above 2 m persisted at the Cairns waverider buoy from 13 to 15 December. Winds strengthened again on 17 December, producing larger-than-normal waves for about 24 h (Hs peaking at just below 2 m and Hmax around 4 m).

## Methods

Three-dimensional elevation models derived from drone imagery and RTK-GPS surveys were used to quantify the geomorphic changes at Low Island following TC Jasper.

### Generation of 3D elevation models

#### Drone imagery acquisition

Three aerial drone surveys were conducted at Low Isles using DJI Mavic-3 (June 2023, June 2024) and Phantom 4 Pro RTK (January 2024) drones. Survey details including flight altitude, use of ground control points (GCPs), ground resolution and survey error are summarised in Supplementary Table 1.

#### RTK-GPS surveys

Ground surveys were conducted on Low Island to georectify drone imagery and ensure digital elevation model (DEM) accuracy. These surveys used a Trimble real-time kinematic GPS (RTK-GPS) R8 base and rover. The base station operated for over six hours daily to maximise positional accuracy. Base station data were submitted to the Australian Government GPS Processing Service (AUSPOS) to correct rover-collected survey points and standardise all RTK-GPS data, achieving an accuracy of 0.014 m (x), 0.015 m (y), and 0.051 m in ellipsoidal height. Key features were mapped including the cay "toe of beach" (TOB), vegetation line, and beachrock outcrops using continuous topographic point storage (Dawson, 2021; Talavera et al., 2021) (Figure 5). The TOB is the seaward edge of the island where the beach meets the near horizontal reef flat, often delineated by a distinct break in slope and/or change from coarser beach sediment to finer rippled sediment on the reef flat (Smithers and Hoeke, 2014; Boak and Turner, 2005). Additionally, the location and elevation of each GCP were established using the RTK-GPS rover unit for processing of the drone images to generate orthomosaics and DEMs (Joyce et al., 2018). One hundred ground validation points were taken at randomly distributed locations across the island and proximal reef flat using the validation point storage setting.

#### Data Preparation

All photographs and orthomosaics were aligned into the same coordinate system and georeferenced using tools within ArcGIS Pro version 3.2.1. and Agisoft Metashape version 2.1.3. The drone imagery was processed in ArcGIS Pro in a separate Ortho Mapping Workspace and Agisoft Metashape to generate orthomosaics. To correct the horizontal and vertical dimensions of the orthomosaic (Joyce et al., 2018), GCP survey points collected using RTK-GPS were imported into Ortho Mapping Workspace as a point file. Each point was then manually matched to the centre of the corresponding GCPs using the *manage GCPs* tool.

### Elevation and volume change analysis

Change in cay volume between the three elevation datasets was calculated using DEMs constructed with the same coordinate system and corrected to a common elevation datum using processed RTK-GPS data (Joyce et al., 2018; Hamylton et al., 2019). Digital surface models (DSM) followed by DEMs were constructed in the ArcGIS Pro Ortho Mapping Workspace. An area of interest (AOI) was extracted using the Extract by Mask tool to exclude parts of the DEM underwater or vegetated as these areas did not return reliable data.

The Raster Calculator was used to measure pixel-by-pixel elevation change between the June 2023 and January 2024 DEMs, representing beach changes before and after TC Jasper. Net volume change was calculated using the Surface Volume Tool in ArcGIS Pro, with the TOB elevation (1.35 m LAT) as the base reference. The June 2024 DEM was excluded due to limited survey coverage from higher tides. To assess sediment movement, the shoreline was divided into four compartments based on location and geomorphic features: northern shoreline (1); eastern shoreline and spit (2); southern shoreline (3); and the western shoreline and spit (4). Volume changes were calculated per compartment to quantify net sediment transfers, and total beach volume change was derived by summing all compartments.

### Monthly shoreline analysis using Sentinel-2 imagery

The drone imagery used to produce the DEMs used in this investigation was captured approximately 6 months before and after TC Jasper. Therefore, geomorphological differences established by comparing these DEMS alone cannot be fully attributed to cyclone impacts. To address this challenge, the average monthly shoreline position of Low Island was established from Sentinel-2 Level 2A imagery (available for Low Isles since 2017, averages calculated from this dataset) and compared with the same months before and after TC Jasper. This approach assumes that the differences between the average monthly and monthly post-TC Jasper shorelines reflect the cyclone's impact. The monthly average shoreline positions were established by selecting the Sentinel-2 image for each month between January 2017 and April 2024. Although coarse (10 m pixel resolution), Lazarus et al. (2025) showed Sentinel-2 imagery can reliably be used to detect the TOB of even the smallest GBR cays, irrespective of tide. Images with less than 30% cloud cover and low sun glint were prioritised. In ArcGIS Pro, 50 points were manually placed around the island's TOB to represent the shoreline position. Using the *Calculate Geometry* tool, the coordinates of each shoreline point were calculated and saved as x and y values in the attribute table. The x and y values for each of the 50 points were averaged across the available years to produce a monthly mean shoreline. Monthly averages were re-imported into ArcGIS Pro to visualise changes in average shoreline position for each month between 2017 to 2024. Monthly shoreline positions at Low Island for the period between June 2023 and June 2024 were established using the same approach (without the eight-year averaging). The shoreline records generated by this procedure enable shoreline positions and changes for the five months prior and post-TC Jasper to be assessed.

## Results

Low Island markedly changed shape between the June 2023, January 2024 and June 2024 surveys (Figure 3), as beach sediments were redistributed around the cay's shoreline. Shorelines derived from Sentinel-2 imagery captured 11 days before and 14 days after the passage of TC Jasper suggest significant shoreline changes were associated with this event (Figure 5c).

### Geomorphic changes associated with tropical cyclone Jasper

#### June 2023 and January 2024: Cyclone effects

Comparison of the elevation models from June 2023 and January 2024 revealed changes to Low Island's shoreline morphology over this interval. Key changes include erosion along the northern

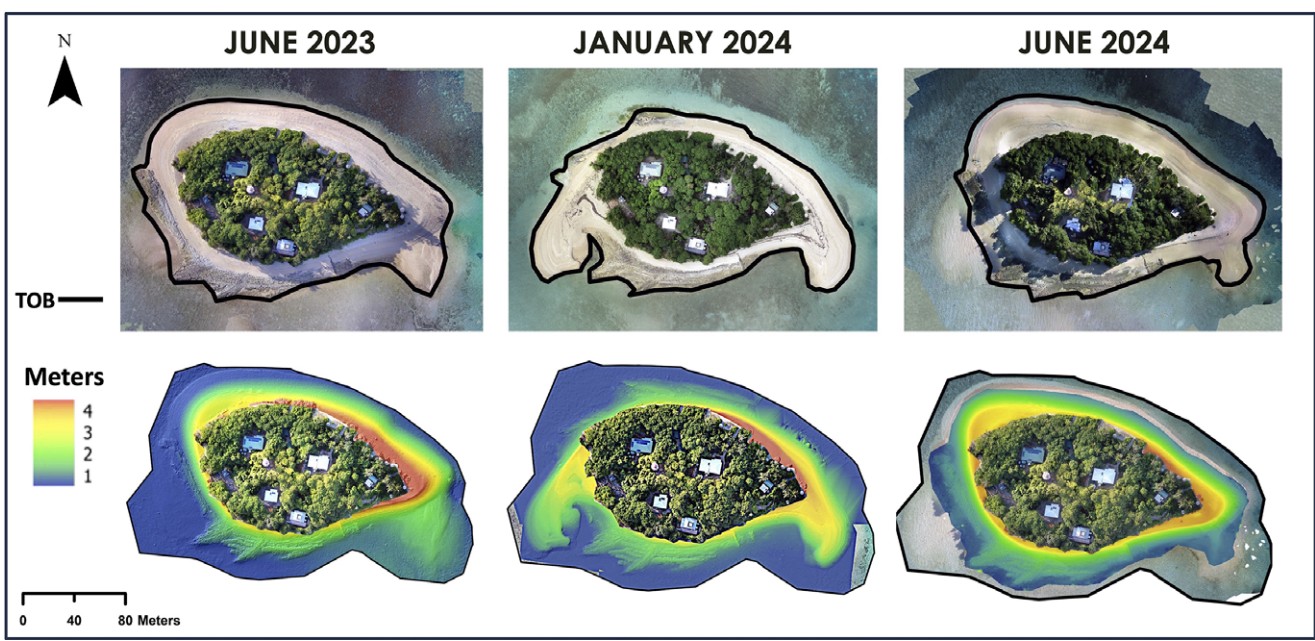

**Figure 3.** Orthomosaics and DEMs generated from field surveys. Toe of beach traces are defined using DEMs.

shoreline, particularly in the northwest corner, where extensive beach rock outcrops were exposed. Two large spits also formed at the eastern and western ends of the cay, both extending southwards over the reef flat (Figure 4.1).

The largest loss of volume (3,245 m$^3$) occurred on the northern cay shoreline (compartment 1) (Figure 4.1a). Erosion at the western end of compartment 1 resulted in retreat of the beach and loss of the berm, reducing the elevation in this area by as much as 1.9 m. Sand volume at compartment 2 at the eastern end of the cay decreased by 1,743 m$^3$. However, the elevation change model shows elevation gains of up to 1.5 m in some areas of compartment 2, with the spit extending approximately 51 m across the reef flat. The southern shoreline (compartment 3) is dominated by beachrock. The volume of this compartment changed little (net loss just 188 m$^3$). The largest changes occurred in compartment 4 at the southwest end of the cay. Volume increased by 4,342 m$^3$ and elevations by as much as 1.6–2.1 m where a large southward curving spit formed. This western spit extends 73 m over the reef flat beyond the shoreline mapped in June 2023, covering an area of around 2,740 m$^2$ and including almost 2,845 m$^3$ of sediment. The volume of sediment lost from all the compartments combined between June 2023 and January 2024 was very small – just 186 m$^3$ (Figure 4.1a).

### *January 2024 – June 2024: Post-cyclone recovery*

The June 2024 survey provides insight into Low Island's recovery six months after TC Jasper. The elevation change model showed accretion on the northern shoreline (Compartment 1) and the largest elevation gain (3 m) at the northwestern corner (Figure 4.2c). These trends in sediment movement are the opposite of those observed following TC Jasper. The elevation of the southern edges of both the west and eastern spits (Compartments 2 and 4) was reduced, likely due to sediment being moved back towards the cay with the trade-wind-driven waves.

### *June 2023 – June 2024: A return to the original shoreline position*

A comparison of the DEMs from June 2023 and June 2024 indicates that the beaches of Low Island had almost completely recovered to their pre-cyclone morphology and position just six months after TC Jasper. Overall, elevation differences between the June 2023 and June 2024 elevation models were relatively small, with a general trend of erosion along the eastern shoreline and accretion along the western and northwestern shoreline. Compartments 3 and 4, on the southwestern side of the island, changed least, with elevation differences of only ±0.5 m compared to the June 2023 survey. In contrast, the largest elevation changes were observed in Compartment 1 where the northwestern corner vertically accreted by 1.4–1.6 m and the northeastern shoreline eroded by 0.9–1.2 m (Figure 4.3e).

### *Historical and seasonal context of Low Island's shoreline*

The average monthly shorelines from 2017 to 2024 show relatively small seasonal shifts in Low Island's shoreline, with elongated eastern and western spits visible in February and March but not in January (Figure 5). Shorelines mapped for June 2023 to June 2024 show these spits appearing in December 2023. Sentinel-2 images were captured on 2nd and 27th December, just before and after TC Jasper passed close to Low Isles on December 13 (Figure 5). The pre-TC Jasper images show that the northern shoreline was uneroded, and eastern and western spits had not formed by 2nd December. However, the images captured on 27th December record both erosion of the northern shoreline and significant sediment redistribution to form conspicuous spits at the eastern and western ends of the cay. Energetic hydrodynamic conditions during TC Jasper are implicated as the cause of observed shoreline changes by this coincident timing.

### Discussion

### *Quantifying morphological and volumetric shoreline changes following a severe storm*

Low Island's response to TC Jasper demonstrates the sensitivity of small cays to hydrodynamic conditions that differ from prevailing conditions, with sediment redistributed across multiple shoreline

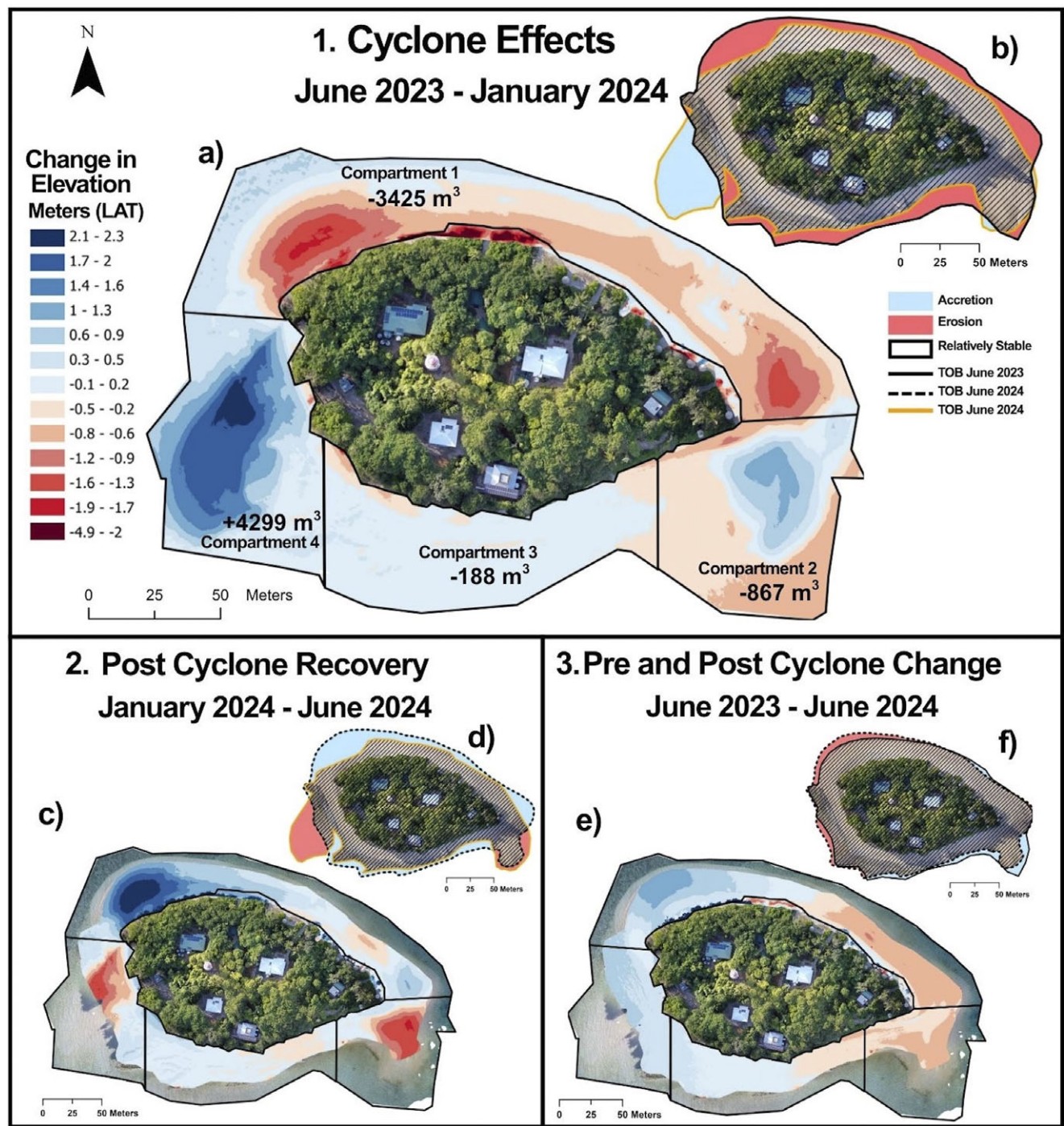

**Figure 4.** Volumetric and planimetric changes measured at Low Island across three periods to assess the effects of and recovery after tropical cyclone Jasper. (a, c, e) Elevation change models derived from the drone imagery captured pre- and post-cyclone. In net terms, dark red areas have undergone the largest volume changes, measured in reductions in elevation (erosion depth), and dark blue areas have built up the most (vertically accreted) over this period. Values in (a) indicate net change in sediment volume for each compartment; note: up or down arrows reflect this increase or decrease in sediment volume, not direction of transport. (b, d, f) Planimetric shoreline changes of the cay's toe of beach. Blue colour indicates shoreline accretion, red; erosion, and hatched; relative stability.

compartments. Six weeks after TC Jasper, the upper northern beach remained steeply scarped. The presence of steep scarping suggests two key points: (1) sediments from the upper beach that are normally above the action of prevailing waves and currents were redistributed during TC Jasper and (2) morphological recovery of the upper beach is progressing more slowly than on the lower beach. At the northwestern corner of Low Island, the shoreline retreated up to 30 m during TC Jasper, producing erosion scarps 0.5–1.2 m high at the vegetation line. In contrast, the southern shoreline between the spits was not directly exposed to high wave action during TC Jasper and was largely unchanged.

Three key lines of evidence reinforce the interpretation of the northern beach erosion being sediment reworking and redistribution around the cay, rather than net export from the system: (1) the small

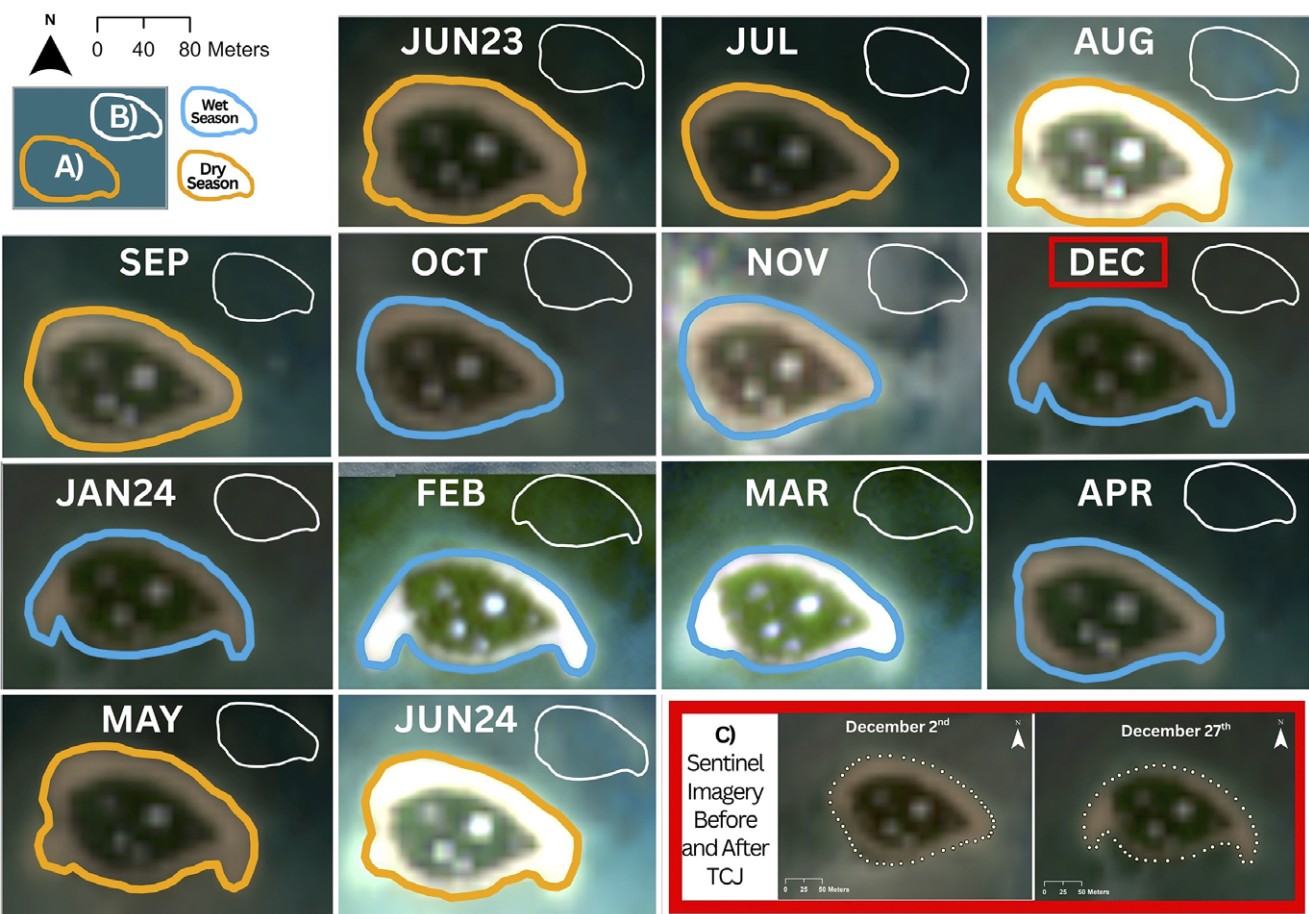

**Figure 5.** Comparison of monthly shorelines of Low Island during the cyclone year (a: June 2023 to June 2024) with the average shoreline position over the previous eight years (b: 2017–2024). (c) Satellite images from December 2nd and 27th, showing the shoreline before and after tropical cyclone Jasper. Shorelines were digitised using Sentinel-2 satellite imagery. Sentinel-2 images were not available February and March 2024, images for these months are instead from Planet Labs Satellite Imagery.

net sediment loss quantified in the volumetric assessment following TC Jasper; (2) the elongation of the eastern and western spits towards the south; and (3) field observations of fresh sediment fans deposited above many of the conspicuous erosion scarps on the northern shoreline, indicating onshore sediment transport by waves. The large amount of sediment moved to create new features, including the erosion scarps and extensive spits, over just a few days demonstrates the sensitivity of cay morphology to higher-than-average wave energy. Many studies have similarly demonstrated that cyclones, as isolated events, can perform considerable geomorphic work on low-lying reef islands over very short timescales (Smithers and Hoeke, 2014). These observations are particularly timely as rising sea levels alter reef-top hydrodynamics, and a change in the frequency and intensity of TCs in a changing climate could increase cay morphodynamics.

The June 2024 survey completed approximately 6 months after TC Jasper offers valuable insights into Low Island's geomorphological recovery. Although significant geomorphological change occurred rapidly during TC Jasper, comparison of the January and June 2024 DEMs (Figures 3 and 4) shows sediment moving from the eastern and western spits back towards the cay's beaches, and towards Low Island's northern shoreline, in the months

following the event. The relatively rapid post-TC Jasper recovery of the shoreline morphology at Low Island under prevailing conditions is not unprecedented. Steers and Kemp (1937) reported that Low Island had "practically regained its former shape" within two years of the 1934 cyclone which at Low Isles produced a storm surge conservatively estimated as approaching 2 m and significantly modified the surrounding reef (Moorehouse, 1936 – see later discussion). A comparison of the elevation models created for June 2023 and June 2024 (Figure 3) also shows that Low Island was very similar in size, shape and elevation in June before and after TC Jasper, suggesting the observed morphology is potentially the cay's modal "winter state" and again emphasising the rapid recovery to this modal form possible under prevailing hydrodynamic conditions.

### A comparison with seasonal shoreline dynamics and historical events

Understanding the relative influence of cyclone-driven events compared to seasonal shoreline processes is crucial for interpreting the long-term stability of reef islands, particularly as the frequency of more intense cyclones is projected to increase due to climate change

(IPCC, 2023). Here, the effects of TC Jasper on Low Island are discussed and compared with seasonal and longer-term shoreline changes inferred from Sentinel-2 imagery, historical records and reports of previous cyclones.

### Seasonal shoreline dynamics

Seasonal shoreline changes at Low Island have been previously documented by Stoddart et al. (1978) and by Hamylton et al. (2019). Both described variations in sand deposition around the cay which coincided with seasonal changes in wind and wave directions. Similar changes to cay shape, particularly shifts in beach sediments and associated spits, are well documented elsewhere (Kench and Mann, 2017; Mann and Westphal, 2014), with studies on the GBR showing that prevailing winds and incident waves drive sediment movement around reef island shorelines (Flood, 1986; Hamylton and Puotinen, 2015). Although these seasonal sediment shifts occur, and may vary interannually in magnitude, over longer timescales, they are generally balanced, with net island area and sediment volume remaining relatively stable (Kench and Brander, 2006; Dawson and Smithers, 2010; Costa et al., 2019). This is certainly the case at Low Island, where comparisons with Spender's (Spender, 1930) detailed maps (1928–1929) show that, after 45 years (Stoddart et al., 1978), and 90 years (Hamylton et al., 2019), the cay's area has changed by less than 1%.

### Historical cyclone impacts: The 1934 cyclone event

Many severe cyclones have passed close to Low Isles in the past, including a system in 1934 for which detailed descriptions of the geomorphic impacts are available (Moorehouse, 1936). This cyclone made landfall near Cape Tribulation on March 12 with a reported barometric pressure of 978, corresponding to a Category 2 system (Callaghan et al., 2020). The event caused widespread destruction from storm tides and high winds. A storm tide of 9.1 m was reported at nearby Baileys Creek and a storm surge of 1.8 m was recorded at Port Douglas (Freeman et al., 2020). The cyclone struck Low Isles on 11 March, 1934, with Moorehouse (1936, p. 40) reporting that Low Island was "considerably altered" by the event, with "much sand being taken from the windward [(eastern)] sides and deposited on the western or lee shore". The 1934 cyclone and TC Jasper 2023 appear to have been of similar intensity and track when passing Low Isles, and they produced very similar geomorphological outcomes.

### Deciphering event-driven vs. seasonal change

Interpreting the impacts of TC Jasper within the context of historical high-energy events and seasonal shoreline changes enables us to disentangle short-term storm impacts from longer-term coastal change. Comparison with the 1934 cyclone clearly demonstrates how high-energy events cause large-scale, abrupt changes, while longer-term satellite observations reveal gradual sediment deposition and reworking that enable the cay's recovery between events. Seasonal variations in hydrodynamics, combined with episodic disturbances and tidal and El Niño–Southern Oscillation (ENSO)-related variability, drive the cay's flux between phases of major change and slower rebuilding. Prevailing southeasterly trade winds during the dry season (May–October) gradually transport sediment to the northern side of the cay, building broader beaches and berms, whereas cyclones rapidly redistribute sediment from the exposed to more protected shores. We note that the shoreline exposed to higher wave energy may differ from that more usually affected by prevailing wind waves. During TC Jasper, erosion, scarping and onshore sand lobe deposition indicate waves approached from a more northerly bearing onto a shoreline that is normally well protected from the prevailing southeasterly wind waves. Although these waves are not exceptionally larger than those experienced windward of Low Isles during peak southeasterly trade winds (Hmax 4 m, Hmax 2 m), they are considerably bigger than those normally experienced on the sheltered northern shoreline. Elevated depths due to the storm surge would also allow these waves to work for longer and higher on the beach face, increasing the geomorphological impact of this event (Woodroffe, 2008; Smithers and Hoeke, 2014; Bernier et al., 2024).

The curved eastern and western spits towards the south are characteristic features formed by both high-energy events and by wet-season wind and waves which can propagate from the north. Cyclones may accelerate their development or cause them to appear earlier in the season, as suggested by the eight-year average monthly shoreline data, which shows spits forming in February rather than December. Stoddart et al. (1978) noted similar characteristics of these features over the 45-year period from 1929–1975 and concluded "*in part they are doubtless seasonal, in part responses to random storm events*" (p. 76). The Low Isles lighthouse keeper, quoted by Fairbridge and Teichert (1948, p. 74), further attested to the recurrence of such variations "*every year with the change of season.*" Hamylton et al. (2016) observed comparable terminal spits at multiple islands farther south on the GBR, consistent with the broader morphodynamic regime of seasonal and periodic change of reef islands described by Hamylton and Puotinen (2015).

Low Isles is a low-wooded island with a mangrove forest behind storm-wave-deposited shingle ramparts on the windward side of the reef platform. The ramparts and the mangrove forest influence the nature of hydrodynamic processes over the reef platform. Historical descriptions, mapping and imagery indicate the elevation and extent of ramparts and mangroves over the reef platform have changed markedly over the past century, modifying wave energy accessing the cay and potentially the rate and direction of spit reworking (and shoreline recovery) between events. A broad cycle of event and recovery is nonetheless observed at Low Island, with sediment redistributed alongshore as described above.

### Conclusion

Low Island is a dynamic vegetated cay. Sedimentary dynamics associated with both seasonal conditions and high-energy events have driven movements in its shoreline and position on the reef platform in the past (see Moorehouse, 1936; Hamylton et al., 2019; Smithers, 2022) and will continue to do so in the future. These geomorphological patterns express the balance between episodic high-energy cyclone events and prevailing hydrodynamic conditions that drive sediment transport and deposition, where cyclones can redistribute large volumes of sediment around the cay in short time periods (days), but prevailing seasonal conditions can also produce relatively rapid recovery (months to years).

With the uncertainty of a changing climate, the future trajectory of Low Island may not follow past patterns. It is already clear that prevailing environmental conditions and episodic storm events have different geomorphologic implications for Low Island, and both prevailing environmental conditions and frequency of episodic storm events are projected to change. The application of drone imagery and high-resolution 3D elevation models in this study provides new insight into sediment redistribution at event, seasonal and interannual timescales. Such technologies can be applied to quantify volumetric change on cay beaches, improving understandings of reef island dynamics.

**Open peer review.** To view the open peer review materials for this article, please visit http://doi.org/10.1017/cft.2025.10017.

**Supplementary material.** The supplementary material for this article can be found at http://doi.org/10.1017/cft.2025.10017.

**Acknowledgements.** We thank the Queensland Parks and Wildlife Service (QPWS) for logistical support during fieldwork. Permission to operate a drone over Low Island was granted by the Great Barrier Reef Marine Park Authority under Part 5.4 of the Zoning Plan as a management activity.

**Author contribution.** Holly Muecke: Writing – original draft, review and editing, visualisation, validation, methodology, investigation, data collection, conceptualisation. Scott Smithers: Writing – original draft, review and editing, visualisation, supervision, methodology, conceptualisation, data collection. Stephanie Duce: Writing – original draft, review and editing, visualisation, supervision, methodology, investigation, conceptualisation, data collection. Sarah Hamylton: Methodology, investigation, writing – review and editing, supervision, data collection. Emily Lazarus: methodology, investigation, data collection.

**Financial support.** This work was supported by the Hunter Research Grant (HM) and partially funded through the partnership between the Australian Government's Reef Trust and the Great Barrier Reef Foundation as part of the *Integrated Monitoring and Reporting Program – Critical Monitoring Stage 2* (SS & SD) and the Australian Research Council (S.H. and K-L.R., Discovery Project DP210100739).

**Competing interests.** The authors declare no known conflicts of interest that could have influenced the work reported in this paper.

**AI statement.** The author used OpenAI's ChatGPT (version GPT-4, accessed via ChatGPT Free) to assist in the development of R code for visualising wind frequency data as rose plots, plotting the track of cyclone Jasper and creating the wave plots. The tool was used interactively to refine plotting scripts (using `ggplot2` and `tidyverse` packages), ensure appropriate handling of data and improve the clarity and consistency of visual presentation. The AI tool was not used to interpret results. All code was reviewed, tested and adapted by the author to meet the research and publication requirements. Additionally, Open-AI's ChatGPT (version GPT-5-mini, accessed via ChatGPT Free) was used for minor language and grammar edits in the manuscript text; it did not contribute to any interpretation of results or the conclusions presented in the manuscript.

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
