## [Reviewer Report]

Overall comments: This is a nicely conducted study within the constraints of data/image availability. It reports on the effects of what was actually a quite low Cat (2) cyclone on a vegetated sand cay in the northern GBR. The data presented (on sign wave heights etc – which arguably should be the first part of the Results) actually seem to show that significant wave heights during the event were not that much higher than during seasonal events (see Fig 2 v SI Fig 1). Instead, the main difference appears to be direction of wave approach which are different to the seasonal approaches. The net effect of the TC was quite modest with island shoreline readjustments reflected mainly as a slug of sediment moving around to, especially, the SW side. However, the majority of this material seems to have been reworked back in a relatively short period after the event.

Perhaps a key point here is that this was not a major cyclone event (e.g., Cat 4/5). The real take home message to me thus seems like it should be that lower Cat cyclones do geomorphic work, perhaps especially where they impact islands from directions outside those of normal seasonal wave approaches, but that recovery can be quite fast. This nuance does not seem to come through in the impact/abstract but I feel needs to. This maybe then also leads to the need for a stronger discussion of/ comparison to what major TC events have been shown to do to islands (Bebe and others) - so a clearer comparison that these big events can do a lot of geomorphic work and indeed lead to island-building. This would to me give the paper a bit more depth and would more fully support what might be the useful contribution here that minor events are mainly exacerbating or modifying islands slightly outside seasonally shifting shoreline patterns, but that these are recoverable. However, if as some models suggest TC intensity not frequency will increase the implications under climate change may be severe.

Other comments/points.

L19 add year to date please.

L19 “calcium carbonate sands” or gravels or mixtures depending on location.

L21 “exhibit spatial and temporal patterns of localised shoreline change,” reference to Kench & Brander 2006 would be useful here as one good example.

The terms reef island, sand cay and just cay are all used on the first page. I would suggest some consistency would help and explain how a sand cay differs from a reef island.

Abstract might useful also explain how the post-event changes compared to the seasonal changes that usually occur – this is alluded to methodologically, but no finding is summarised. Indeed, coming back to this point now having read the whole paper this is really important to be clear about because 1) the magnitude of change was actually quite small and 2) the event seems to have caused a response more because of the unusual direction of wave approach than the sign wave heights, and 3) most sediment seems to have returned during later seasonal wave exposure.

I think also in the abstract that it would be worth emphasising that this was a Cat 2 event. A reasonable amount of sediment movement seems to have occurred despite this and so the fact that there was a noted geomorphic response is relevant and could be mentioned in terms of what a major event might do.

L50 “the values they support”. Do you mean services, functions, people? Values seems the wrong term here.

L52 “geomorphic impacts”. Could you expand since this is quite a generic term.

L146. Some of the material in here and the plotted data (Fig. 2) seem really to be results not background material to me in the sense that this section is reporting significant wave heights etc during the event. I wonder if this should be moved to the results section. Indeed, there may be merit in that because it would allow a clearer comparison with the normal season wave height/direction data. My reading of this data is that significant wave heights were not really much above those occurring during parts of the wet season, but the main difference was the direction of approach. A revised combination of Fig 2 and SI fig 1 may be useful here. I feel this is important and should perhaps be emphasised more strongly in e.g., abstract. Sign wave height is important but also direction of approach. Thus, during high energy events, even relatively modest (Cat 2) ones the different direction of wave approach can lead to meaningful sediment remobilisation. On L330 the point is made “when exposed to higher-than-average wave energy”. Actually as noted above wave heights were not (to my reading) much above those that are shown in SI fig 1 – so again it seems that it is perhaps the direction of high wave approach that is more important here. Significantly, and this is supported by later parts of the discussion, the amount of actual long-term change to the main part of the island “structure” seems quite limited – these lower magnitude cyclone events are either exacerbating season events or causing short-term sediment transfer of material from the beaches but this neutralises over quite short periods when the season winds move.

L228 “try to tease out”, and L232 “may have been brought about by the cyclone”. This concerningly now all sounds less certain in terms of event attribution than implied earlier in the paper.

Fig 3 – the image for June 2024 appears actually to still show the same body of sediment sitting to the SW of the island. This was traced around in Jan 2024 but not in June 2024. To me it seems a slight matter of interpretation about whether this body of moved sand has actually gone. This feeds through into Fig 4. I know the interpretation is based on toe of beach surveys but actually, and although perhaps now sitting a a very slightly lower elevation, that body of sediment in the SW block is still there (Fig. 3 – June 2024)

Fig 4 – as someone who is red-green colour blind this colour scheme is not easy to interpret. The authors might think about perhaps a blue to green scheme.

L414 “Low Island is a dynamic unvegetated cay”. I am confused. I assume you mean vegetated and that the peripheral margins are dynamic?

---

## [Reviewer Report]

The manuscript titled “Effects of Tropical Cyclone Jasper at Low Island, Northern Great Barrier Reef” presents a timely and relevant case study of storm impacts on reef islands. The study is interesting and overall well-written, with clear field and remote sensing data collection and strong potential to contribute to our understanding of how unconsolidated reef islands respond to cyclonic events. This manuscript presents a valuable and well-executed dataset, and the results—particularly the documentation of shoreline and sediment reorganization without major net loss—are important for coastal geomorphology and island resilience research.

However, the manuscript in its current form requires major revision before it can be considered for publication in Coastal Futures. The abstract and introduction are at times too superficial and informal, lacking quantitative context and broader scientific framing beyond Low Isles papers. The study objectives are presented in a way that feels too site-specific and method-oriented, rather than highlighting the transferrable scientific contributions to reef island geomorphology worldwide. In fact, specific objective 2 is questionably an objective achieved by this study (it was not a dense discussion… it was good but not to be an objective of the paper). The results are promising and supported by clever figures (e.g., Figure 5), but the description is often too brief, missing detail/depth. Similarly, the discussion is somewhat repetitive (in relation to results) and site-focused.

In summary, this paper provides an important dataset and addresses a significant event, but revisions are necessary to strengthen the framing, add substance to the results and discussion, and elevate the study from a descriptive case study of Low Isles to a contribution with broader relevance. Below I provide more specific comments to support the improvement of the manuscript.

Specific comments

Line 19: Please clarify — Dec 2–13 of which year?

Line 27: This seems to be intended as the paper’s objective, but it could be improved. As the authors highlight in the importance statement, the focus should be more on island responses to storm activity rather than simply “examining shoreline changes in Low Isles.” While I personally find Low Isles interesting, other readers may not, and the objective needs to be framed more in terms of advancing scientific understanding rather than describing a single case study.

Line 32–33: Results are said to be compared to seasonal shoreline changes… but then what? Could the authors provide a bit more detail about their findings here?

Line 52: “impacts of high energy storm events”.. after this maybe add “on small unconsolidated reef islands”?

Line 54–58: This section reads as very basic TC background. I suggest using this space instead to build a stronger case for TC impacts on reef systems: e.g., coral breakage, sediment generation, sediment transport, and subsequent island morphological change.

Line 66–68: This is a long sentence. It is understandable, but worth revising for clarity.

Line 70–72: The authors touch briefly here on what I mentioned above, but I suggest opening the paragraph with this point. Also, a citation is needed.

Line 85: ENSO should be defined on first use: El Niño–Southern Oscillation (ENSO). The paragraph is acceptable, but it reads more like a study area description than the close of the introduction.

Line 90–97: This section ties back to my earlier comment about the objective being too site-specific. The objective does not have to be limited to Low Isles; it can be reframed for broader relevance. Also, the two listed “specific objectives” read more like methods than scientific objectives.

Abstract and Introduction: The writing is generally fine, but at times it feels informal and somewhat superficial. Stronger, more consistent arguments would help — for example, including quantitative values, case examples from other cyclone events, and clear take-home conclusions rather than simply “previous studies also observed shoreline changes.” This would help build a more compelling narrative that supports later discussion.

Line 99: Low Isles is described as “a low wooded island” — but aren’t there two islands? Please clarify.

Line 140: The sentence about wave periods and heights is confusing and the parentheses messy. Also, “larger wave height” is unclear — please revise.

Line 142: waves are smaller by 4 cm? Hmax is actually higher.

Line 147: I recommend calling TC Jasper instead of TCJ.

Line 158: Use “wave height” instead of “wave sizes.”

Line 160: Use “waverider” instead of “wave-rider.” Please check consistency throughout the manuscript.

Line 164: After introducing “significant wave height,” use the shorthand Hs consistently throughout.

Drone imagery acquisition section: While Table 1 provides some information, this section could benefit from more methodological detail: e.g., number of GCPs, survey error, etc. Also, regarding DEM generation: my understanding is that angled imagery (not strictly nadir/perpendicular) is required. If perpendicular imagery was used for a specific reason, please explain. If I’m mistaken, disregard this comment.

Line 213: The Talavera citation is unclear here. Also, ensure consistent naming of TC Jasper throughout.

Line 239–241: The Excel processing description is unnecessary. I suggest removing the sentence about grouping monthly files in Excel and starting instead with: “The x and y values for each of the 50 points were averaged…”

Line 252: Agreed — and the figures (which should be cited more often in the text) present very interesting results. I recommend merging this short paragraph with the previous one.

Line 258: occurred due to? TC Jasper. Also, Figure 7?

Line 282: The finding that sediment was rearranged across island compartments with minimal net loss is very interesting.

Jan–Jun 24 and Jun 23–Jun 24 sections: These deserve a more detailed description, with values presented as in the Jun 23–Jan 24 section. Avoid superficial descriptions.

Figure 5: This is a clever and effective figure — I really like it. However, as noted above, the accompanying results section (3.2) could provide more detailed interpretation.

Section 4.1 title: Consider revising to something more like “Morphological and volumetric changes following a severe storm”.

Line 311–321: This entire paragraph reads more like results. Given the short length of the paper, it feels repetitive. At the same time, I missed some of this detail while reading the actual results section. Also, the discussion of Jasper’s storm waves (height, direction), the track crossing the reef and exposing Low Isles, and the resulting morphology changes should be expanded here.

Line 329: Delete “short” — “just a few days” is clearer.

Line 330: “Many studies” — please cite specific examples to strengthen the discussion. Without references, the manuscript risks reading as a site-specific report rather than a contribution of broader relevance.

Line 333: The statement about an “increase in frequency and intensity of storms” needs a citation. Note that evidence for this is not uniform: for example, Chand et al. (2019) found statistically significant decreasing trends in both seasonal TC frequency (∼−0.115 year−1) and severe TC frequency (∼−0.112 year−1) over Australia, with >95% confidence. https://doi.org/10.1002/wcc.602

Line 375: Please correct the citation year.

Line 378: Consider merging sections to avoid so many breaks — the current structure interrupts the flow.

Line 406: While the mangroves on Low Wooded Island clearly protect Low Island from SE waves, the spits appear to be a response to diffraction of high NE waves around the island.

---

## [Editor Report]

Dear authors,

Both reviewers have recommended a major revision of your manuscript. After re-reading your paper alongside the reviewers’ comments, I agree with their recommendation. The feedback provided is constructive and will improve the clarity and discussion of an already very interesting paper. 

One particular suggestion is to contextualize the wave height of the tropical cyclone in comparison with seasonal storms, which will aid in the interpretation of your results. They also note that the manuscript can use more discussion of the results. 

I look forward to receiving your revised manuscript.

---

## [Reviewer Report]

I feel that the authors have done a good job of responding to the reviews and that the M/s is now reading very well.

---

## [Editor Report]

Reviewer 1 recommended acceptance of the manuscript. Reviewer 2 declined the second review.

I have reviewed the authors’ responses to both reviewers and examined the revised manuscript. The authors addressed all the comments (major and minor) raised by both reviewers. After re-reading the paper, I have no additional comments, concerns, or suggestions. I therefore recommend that the paper be accepted for publication.